# Development and Validation of an Auricular Acupuncture Protocol for the Management of Chemotherapy-Induced Nausea and Vomiting in Cancer Patients

**DOI:** 10.3390/healthcare12020218

**Published:** 2024-01-16

**Authors:** Eliza Mara das Chagas Paiva, Caroline de Castro Moura, Denismar Alves Nogueira, Ana Cláudia Mesquita Garcia

**Affiliations:** 1Interdisciplinary Center for Studies in Palliative Care, Nursing School, Federal University of Alfenas, Alfenas 37130-001, Minas Gerais, Brazil; eliza.paiva@sou.unifal-mg.edu.br (E.M.d.C.P.); caroline.d.moura@ufv.br (C.d.C.M.); denismar.nogueira@unifal-mg.edu.br (D.A.N.); 2Department of Medicine and Nursing, Federal University of Viçosa, Viçosa 36570-900, Minas Gerais, Brazil; 3Institute of Exact Sciences, Federal University of Alfenas, Alfenas 37130-001, Minas Gerais, Brazil; 4Interdisciplinary Cooperation for Ayahuasca Research and Outreach (ICARO), School of Medical Sciences, University of Campinas, Campinas 13083-970, São Paulo, Brazil

**Keywords:** auricular acupuncture, nausea, vomiting, oncology, palliative care, content validity, development and evaluation

## Abstract

Auricular acupuncture (AA) has been used to manage chemotherapy-induced nausea and vomiting (CINV). However, the application of the technique varies widely among the clinical trials that test its effectiveness. The aim of the present study was to develop and clinically validate an AA protocol for the management of CINV in cancer patients. This study was carried out in two stages: (1) development of the AA protocol for the management of CINV and (2) clinical validation of the protocol. The content validity of the protocol was determined by a panel of specialists, with an agreement rate ranging from 85.7% to 100%. In the clinical validation, when administered to cancer patients, the protocol developed has been shown to reduce the incidence, frequency, severity, and length of nausea and vomiting following chemotherapy, as well as the severity of nausea and anticipatory nausea following chemotherapy. This protocol needs to be tested in future studies, including a pilot study with a sham group and a randomized clinical trial, in order to further evaluate its feasibility, acceptability, safety, and clinical usefulness for the management of CINV.

## 1. Introduction

Nausea and vomiting are the main adverse effects in patients undergoing antineoplastic chemotherapy [1]. The incidence of vomiting can reach up to 32.7%; as for nausea, the incidence can reach up to 58% [2]. These manifestations have been associated with a series of complications, such as malnutrition, dehydration, electrolyte imbalance, fatigue, weight loss, anxiety, depression, as well as negative repercussions on social function [3,4]. If not adequately controlled, chemotherapy-induced nausea and vomiting (CINV) can compromise the continuity of cancer treatment and even reduce long-term survival rates [4,5]. 

Due to the complex pathophysiological mechanisms related to CINV, pharmaceutical agents do not always allow for complete remission of these manifestations [2]. Furthermore, in addition to the risk of potential medication interactions with antineoplastic regimens, the excessive use of pharmacological interventions increases the financial burden on patients and public healthcare systems [6]. In light of this, it is worth exploring the role of non-pharmacological methods for the management of CINV as complementary approaches to pharmaceutical agents [7,8,9].

Auricular acupuncture (AA) is a traditional Chinese medicine (TCM) treatment that has been used to treat various health conditions and can be used safely in cancer patients [10,11]. It is a technique in which ear acupoints are stimulated using various devices, such as needles, seeds, spheres, electrostimulation equipment, lasers, cauterization, and magnets, among others [12,13]. It is believed that the stimulation of ear acupoints induces neuromodulation in the central nervous system, based on the stimulation of the trigeminal and vagus nerves [14,15].

Over the last three decades, there has been a substantial increase in the number of publications on the effect of AA for CINV management [16]. Paiva and colleagues [17] carried out a systematic review study with the aim of evaluating the effects of AA in the management of CINV in cancer patients. Although the studies included pointed to the potential benefit of this technique in CINV management, the authors found considerable heterogeneity in the application of the technique among the studies, as well as methodological flaws and a high risk of bias among the clinical trials included in the sample. The evidence therefore remains inconclusive [17]. 

Therefore, considering the evidence-based practice framework for decision making in clinical practice regarding the use of AA in CINV management, it is necessary to produce more robust scientific evidence using adequately validated and standardized protocols [17,18]. Given these considerations, the present study aimed to develop and validate an AA protocol for CINV management in cancer patients undergoing chemotherapy.

## 2. Materials and Methods

This study was carried out following the framework of actions for intervention development proposed by O’Cathain and colleagues [19]. The following actions guided the development of this study’s steps: planning the development process; involving stakeholders, including those who will deliver, use, and benefit from the intervention; composing a team and establish decision-making processes; reviewing published research evidence; drawing upon existing theories; conducting primary data collection; understanding the context; and designing and refining the intervention [19]. Based on this framework, this study was carried out in two stages: (1) development of the AA protocol for the management of CINV and (2) clinical validation of the protocol.

## 3. Development of the AA Protocol for the Management of CINV

### 3.1. Planning the Development Process and Reviewing the Published Scientific Evidence

In line with the recommendations of O’Cathain and colleagues [19], which guide the development of interventions based on scientific evidence for identifying what is relevant and feasible for achieving the intended objectives of the intervention, the AA protocol for the management of CINV used in this study was prepared based on a previously conducted systematic review [17]. 

### 3.2. Drawing upon Existing Theories

According to the results of a systematic review [17], most studies included were based on the theoretical perspective of TCM, which was considered to support the use of AA for the management of CINV. The development of the protocol was also based on the guidelines of the Standards for Reporting Interventions in Controlled Trials of Acupuncture (STRICTA), so that all the details of AA were covered and to favor its use in clinical trials [20]. A map proposed by the World Federation of Acupuncture-Moxibustion Societies (WFAS) was used to help locate and identify the acupoints [21]. The WFAS is a non-governmental international union of acupuncture–moxibustion societies that establishes official relations with the World Health Organization. Its main objective is to promote understanding and cooperation between global acupuncture–moxibustion circles.

### 3.3. Composing a Team and Establishing Decision-Making Processes

The protocol was refined and revised by a committee composed of fourteen specialists with the aim of identifying possible inconsistencies and rectifying them [19]. The number of fourteen specialists was based on the recommendations of Vieira et al. in a study on the validation of nursing protocols [22]. The specialists were instructed to assess the scientific, theoretical, and practical suitability of the AA protocol for CINV management. 

To determine the specialists’ level of professional expertise, a curriculum vitae scoring system was used, which considered the following criteria related to experience and training in the TCM and/or systemic acupuncture and/or AA areas: (1) practical experience of at least four years in the area (mandatory); (2) teaching experience of at least one year in the area; (3) research experience with articles published on the theme in indexed scientific journals; (4) participation of at least two years in a research group in the area; (5) holding a master’s degree with a dissertation in the specific area; (6) holding a PhD with a thesis in the specific area; (7) having completed a course, specialization, or residency in the specific area [23]. For each criterion, one point was added, except for holding a PhD, for which two points were added, in addition to practical experience, for which four points were added. For each year of clinical or teaching experience related to TCM and/or systemic acupuncture and/or AA, an extra point was added [23]. 

Based on the sum of the scores for the established criteria, the specialists who achieved a minimum score of six were invited via email to participate in the protocol validation process [23]. The protocol evaluation tool was sent for analysis using an online form (Google Forms). Each item in the protocol had to be evaluated independently on a 3-point Likert scale, as follows: (1) adequate; (2) partially adequate; (3) inadequate. In addition, the evaluation instrument included blanks in each item, so that the specialists could suggest changes and make remarks and comments about the protocol [24]. 

The processes of judging the protocol’s relevance and the consensus between the specialists were based on the Delphi technique [25]. The suggestions made by the specialists were analyzed using the percentage of absolute agreement, using the following formula: % agreement = (number of participants who agree/total number of participants) × 100. Items with agreement rates equal to or greater than 80% were considered adequate [26]. 

Discussions were held between the research team on the items that failed to reach the established level of agreement, and the suggestions that were deemed relevant were accepted. When the comments presented by the specialists were not considered adequate by the research team, justifications were sent to the specialists for not modifying the items, and a new judgment of the specialists’ opinions was requested, with the aim of reducing the level of disagreement [25]. This process was repeated for three rounds until the established index of agreement had been reached [25].

## 4. Clinical Validation of the AA Protocol for the Management of CINV

The clinical validation was designed to determine the usability and feasibility of the initial version of the AA protocol for CINV among stakeholders. For this purpose, the preliminary application of an intervention in the target population can be based on pre- and post-test measurements [19]. Based on clinical validation, it is possible to identify the need for adjustments to the protocol, in addition to those indicated by the specialists, before conducting larger studies [19]. It is also possible to predict whether the intervention has the potential to be effective in a future implementation in the clinical setting, in a way that is safe for patients [19].

### 4.1. Involving Stakeholders, including Those Who Will Deliver, Use, and Benefit from the Intervention and Carry out Primary Data Collection

#### 4.1.1. Study Design 

A quasi-experimental study was used for the clinical validation of the AA protocol for CINV management.

#### 4.1.2. Sample and Eligibility Criteria

The following inclusion criteria were considered: (1) aged 18 or over, regardless of sex; (2) having a confirmed diagnosis of cancer, of any histological type and disease staging; (3) being on chemotherapy treatment for adjuvant, neoadjuvant, or palliative purposes during the study’s data collection period; (4) having reported nausea and/or vomiting; (5) patients undergoing chemotherapy cycles at intervals of more than one week, such as fortnightly, 21 days or monthly; (6) having time available to participate in the AA sessions; and (7) having a performance status equal to or less than three according to the Eastern Cooperative Oncology Group (ECOG) Performance Status scale [27,28]. 

The ECOG is a performance status assessment scale that aims to verify whether a patient is fit to receive a treatment. Researchers worldwide consider the ECOG Performance Status scale when planning clinical trials to study novel cancer treatments. Patients with high performance status tend to be at greater risk of chemotherapy toxicity and poor outcomes when undergoing treatment [28]. A score of 0 means that the person is fully active, able to perform all their activities without restrictions; a score of 1 is for people who are restricted from performing strenuous activities, but are able to walk and perform light activities, such as housework and office work; a score of 2 characterizes people who are unable to carry out work-related activities, but can walk and perform self-care; a score of 3 indicates limited self-care and means the person is chair- or bed-bound for more than 50% of the time they spend awake; a score of 4 is for people who are totally bed-bound and unable to perform self-care; and a score of 5 represents death [27]. This justifies excluding patients with an ECOG status higher than 3. These patients may face difficulties in complying with the study requirements, such as attending scheduled sessions. In addition, they may present a higher risk of adverse events. Another important factor is that these patients may be less able to provide informed consent and respond to the data collection forms.

Exclusion criteria were as follows: (1) inflammation, infection, ulceration [29], or ear deformities [6]; (2) hearing aid use; (3) confirmed or suspected pregnancy [30]; (4) allergy to micropore^®^ [29].

The criteria for discontinuing the intervention were the following: (1) interruption of the chemotherapy regimen, regardless of the reason; (2) a request from the patient themselves or their legal representative to leave the study; (3) missing AA sessions more than twice, whether consecutively or not [13]; (4) discomfort at the AA site [13].

A sample of 20 patients was established, based on a study previously carried out on the validation of an AA clinical protocol [31].

### 4.2. Recruitment

The patients were recruited between January and July 2023. Before undergoing chemotherapy, the patients were individually asked about the occurrence of nausea and/or vomiting during a nursing visit. When these signs and/or symptoms were reported, the patients were invited to voluntarily participate in the study. 

### 4.3. Instruments

Participants’ sociodemographic and clinical characterization questionnaire consisted of items covering sex; marital status; race/color; frequency of chemotherapy sessions; duration of chemotherapy treatment; and type of cancer. The information was obtained from the patients’ medical records before the intervention was carried out. 

A Visual Numeric Scale (VNS) was used to assess the intensity of nausea and vomiting. It is an instrument with evenly spaced numerical scores (0 to 10), where the higher the number, the greater the intensity of nausea and/or vomiting [32].

The Morrow Assessment of Nausea and Emesis (MANE) scale was to assess CINV; the translated version of the MANE scale adapted for the Brazilian context [33] was used [34]. The scale consists of 16 items that assess the incidence, frequency, severity, and duration of nausea and vomiting in the anticipatory, acute, and late stages, allowing an assessment of these symptoms as distinct phenomena [35]. The instrument’s items are structured as follows: incidence of nausea and vomiting in one or both periods (yes, no); frequency (in times); severity (very weak, weak, moderate, strong, very severe, and unbearable); duration (in hours); the period during which patients experienced the most severe nausea/vomiting (during treatment; 0 to 4 h after treatment; 4 to 8 h after treatment; 8 to 12 h after treatment; 12 to 24 h after treatment; 24 h or more after treatment; never); and how many hours before chemotherapy the first instance of nausea or vomiting occurred [35]. The results are calculated using a Likert scale, which means that each item must be analyzed individually and interpreted according to the individuality of each patient.

### 4.4. Intervention

The patients received AA according to the version of the protocol that had been validated by the specialists in step 1 of this study. In order to ensure operational consistency, the technique was performed by a single nurse with training in systemic and auricular acupuncture and over four years of practical experience in the field [36]. During the intervention, all participants continued to receive the routine antiemetic treatments and medications of the oncology service where the study was carried out.

The patients received five weekly sessions of AA with vaccaria seeds on acupoints related to the management of CINV, including Shénmén (TF4) shenmen; Jiaogan (AH6a) sympathetic nerve; Wèi (CO4) stomach; Pí (CO13) spleen; Gän (CO12) liver; Pízhìxià (AT4) subcortex; and bënmén (C03) cardia (Figure 1). To aid in locating the points, an electric acupoint detector and a standard Chinese auricular map were used [21]. The points were applied unilaterally, starting with the left ear, alternating with each new session. The seeds were attached to the acupoints using hypoallergenic tape. 

Manual acupressure was applied to the points by the interventionist for approximately 30 s on each point, or until the patient reported a feeling of slight discomfort or pain [29,37]. Participants were instructed to apply pressure to each point daily for 30 s, at three times of the day: in the morning, afternoon, and evening, and when feeling nauseous or vomiting [38]. Each participant was also asked to demonstrate acupressure to the interventionist in order to ensure that they had understood the guidelines and would be able to stimulate the points correctly. The process of carrying out the intervention is presented in detail as Appendix A.

### 4.5. Data Collection Procedures

The data were collected at an oncology center in the state of Minas Gerais, in the southeastern region of Brazil, which provides treatment to more than 26 cities in the region. Initially, the participants’ characterization data were collected from their medical records. The MANE scale was then applied through an interview. The participants then received five sessions of AA. At the end of the fifth session, the final and follow-up evaluations were carried out using the MANE scale.

The AA and questionnaires were administered in a private room. To avoid disrupting the service dynamics of the sector where the study was carried out, a single researcher was responsible for evaluating the results and for administering the interventions.

### 4.6. Data Analysis

The IBM SPSS Statistics for Windows 20.0 was used for data analysis with the significance level set as *p* < 0.05. Data normality was verified using the Shapiro–Wilk test. Sociodemographic and clinical data were presented using simple frequencies, measurements of central tendency (mean), and variability (standard deviation—SD). For variables that lacked normal distribution, the median and interquartile range (IQR) values were additionally reported. To compare the effect of the intervention before and after treatment, the McNemar test was used for dichotomous variables, and the Friedman test was used for polytomous variables. 

### 4.7. Ethical Considerations

This study was approved by the Federal University of Alfenas Research Ethics Committee (CAAE: 60933422.0.0000.5142; Number: 5.688.449; 6 October 2022). All participants received written information concerning the research, and all of them granted their informed written consent before participating in the study. No identifying information was recorded within the questionnaire responses collected.

## 5. Results

### 5.1. Development of the AA Protocol for the Management of CINV

The panel of specialists consisted of thirteen specialists from Brazil (93%) and one from the United States (7%), who had between 4 and 27 years of clinical experience in the field of AA, with a mean of 12 years (SD = 7.4) and a median of 10 years (25th percentile = 5.3; 75th percentile = 16.5). In terms of qualifications, 5 (35.7%) held a PhD, 7 (50%) held a master’s degree, and 13 (92.9%) were specialists. The characterization of the specialists is shown in Table 1. 

Consensus on content validity was reached through three rounds of evaluation, when the final analysis of the level of agreement of all the items in the protocol reached an agreement of 85.7% to 100%. Table 2 shows the final version of the proposed protocol (Appendix A) and the results of the content validity assessment. 

### 5.2. Clinical Validation of the AA Protocol for the Management of CINV

The clinical validation of the protocol included the participation of 20 patients, aged between 18 and 79, with a mean age of 58.1 (SD = 15.7) and a median of 59 (25th percentile = 1; 75th percentile = 70) years. The other data characterizing the participants are presented in Table 3.

The mean intensity score of nausea among the patients who participated in the study was 8 (SD = 2.7), with a median of 5 (25th percentile = 5; 75th percentile = 10), while the mean intensity score of vomiting was 4.7 (SD = 3.2), with a median of 5 (25th percentile = 2; 75th percentile = 7.3), according to the visual numeric scale. 

The assessment of CINV in the study participants after applying auricular acupuncture found a significant improvement in the incidence (*p* = 0.004), frequency (*p* = 0.001), severity (*p* = 0.001), and duration of nausea after chemotherapy (*p* = 0.002). Regarding the assessment of vomiting after the last chemotherapy session, a statistically significant difference was found in terms of incidence (*p* = 0.031), frequency (*p* = 0.007), and period (*p* = 0.021). Regarding anticipatory nausea, there was only a reduction in severity (*p* = 0.034) (Table 4).

## 6. Discussion

Evidence-based practice is essential for providing high-quality care to patients, as well as for reducing the variation in protocols in clinical practice [41]. Its three main components are the use of the best evidence currently available, the knowledge and clinical experience of professionals, and the preferences and beliefs of patients [42]. In this context, the implementation of health interventions should be based on these principles, in order to identify the barriers to their applicability in daily clinical practice [43]. In light of this, the purpose of this study was to develop and validate an AA protocol for the management of CINV in cancer patients undergoing chemotherapy, based on evidence from a systematic review, the opinions of a committee of specialists, and, finally, clinical validation involving the stakeholders.

The development of the AA protocol for the management of CINV was based on the solid foundations of TCM. According to the premises of this theoretical framework, CINV represent the clinical manifestation of a pattern of organic disruption caused by chemotherapy and can be interpreted as a strong dose of toxic heat that causes the consumption of Yin energy, especially in the stomach, heart, kidneys, and lungs [44]. It is therefore possible to alleviate these symptoms through TCM treatments, such as AA, which enables the body’s energy to be rebalanced and, consequently, to manage the clinical manifestations [44].

Once a protocol has been developed based on scientific evidence, it is essential to have its content checked by a panel of specialists to ensure that it is relevant and capable of achieving the proposed objectives without posing risks to patients [22,45]. Our study employed a group of specialists who had extensive theoretical training in the field of AA, as well as practical experience, which is key to understanding how the intervention should be adapted for use in clinical practice. In addition, the panel was of a relevant size and included specialists with heterogeneous backgrounds, which allowed the inclusion of feedback and diverse ideas regarding the protocol characteristics, placing them under discussion among the study team [22]. Each item in the protocol was evaluated individually and achieved satisfactory scores in terms of agreement between the specialists [26]. Therefore, it can be assumed that it has satisfactory content validity and can be used in studies involving patients directly. 

More importantly, this study validated the protocol in the reality of clinical practice. The preliminary application of an intervention in a smaller group of patients enables an understanding of its initial suitability, the identification of possible biases, and the necessary adjustments to be made before larger-scale studies are carried out, thus ensuring patient safety [19]. The clinical validation of the AA protocol showed statistically significant results in the reduction in the incidence, frequency, and period of nausea and vomiting after chemotherapy, as well as a reduction in the severity of nausea before and after chemotherapy. These results are in line with the findings of previous clinical trials, which found a statistically significant reduction in CINV after AA treatment [6,30,38,39,46]. In light of the above, it is assumed that the proposed protocol, in principle, does not require any changes and can be considered promising for the treatment of CINV in cancer patients. 

Our protocol suggests the use of AA with seeds, which is the most widely used modality for the treatment of a wide variety of health conditions and in various medical fields, including oncology [36]. This method is low-cost for public healthcare systems and has minimal counterindications and great acceptability among patients, especially when compared to other invasive modalities, such as AA with needles. In addition, it is well accepted among professionals in clinical practice, and AA training courses are relatively short and affordable. These and other aspects make this treatment modality feasible in the context of clinical oncology [47,48]. We suggest a protocol with five AA sessions in weekly intervals, as this is considered sufficient in the literature to treat CINV efficiently in the long term [49]. The protocol recommends keeping the seeds in the ear for a period of seven days, which is considered to be the ideal time for them not to dislodge or come loose, making them safe for the patient [40].

In addition, we suggest the application of seven AA points per session, selected according to their functions, based on the TCM theory, with the aim of treating the syndromic diagnosis as a whole and not just the symptoms in isolation, allowing the body’s energy to be balanced as a whole [4,50]. The Shenmen point has calming, tranquilizing, and general harmonizing effects on the body, including anti-emetic properties [51]. The sympathetic point has the function of regulating the autonomic nerves and can manage nausea and vomiting caused by excitation of the vagus nerve [51]. The stomach point helps in the general regulation of nausea and vomiting and harmonizes the flow of Qi [51]. The spleen point is usually applied in combination with the stomach point, and also works to regulate nausea and vomiting and harmonize Qi [51]. The liver point was selected due to its function of harmonizing this organ, which is also involved in the digestion process, as well as helping to strengthen the spleen, stomach, and regulate Qi [52]. The subcortex point was selected as it helps regulate the excitability of the cerebral cortex and harmonize the gastrointestinal tract [51]. 

This study presents some limitations. Firstly, although the intervention protocol is innovative in terms of including cancer patients for clinical validation, this exercise was carried out in a single oncology center and with a relatively small sample size. Therefore, the results cannot be generalized to other local contexts. Secondly, interference related to expectations with AA treatment could not be tested, as we lacked a control group with sham AA. The protocol that was developed and validated in this study, although promising, comprises only a preliminary stage of a study consisting of several stages to evaluate the effect of AA on the management of CINV. In light of this, more studies are needed, including a pilot study with a sham group, followed by a randomized clinical trial, in order to further evaluate this intervention in terms of feasibility, acceptability, safety, and clinical usefulness for the management of CINV.

## 7. Conclusions

The present study developed and validated an AA protocol for the management of CINV in cancer patients undergoing chemotherapy. This process was based on systematic review evidence and relevant theories and received content validation from a large panel of specialists, in addition to demonstrating positive results in the treatment of CINV in a small group of patients in the clinical validation step. 

The validated version of the protocol provides a treatment course of five AA sessions, held once a week, with vaccaria seeds, according to the principles of TCM, on the Shénmén (TF4) shenmen, Jiaogan (AH6a) sympathetic nerve, Wèi (CO4) stomach, Pí (CO13) spleen, Gän (CO12) liver, Pízhìxià (AT4) subcortex, and bënmén (C03) cardia points, alternating ears at each session. Further studies on the use of the AA protocol for treating CINV are needed in order to monitor its suitability, feasibility, safety, and acceptability in the long term. 

## Figures and Tables

**Figure 1 healthcare-12-00218-f001:**
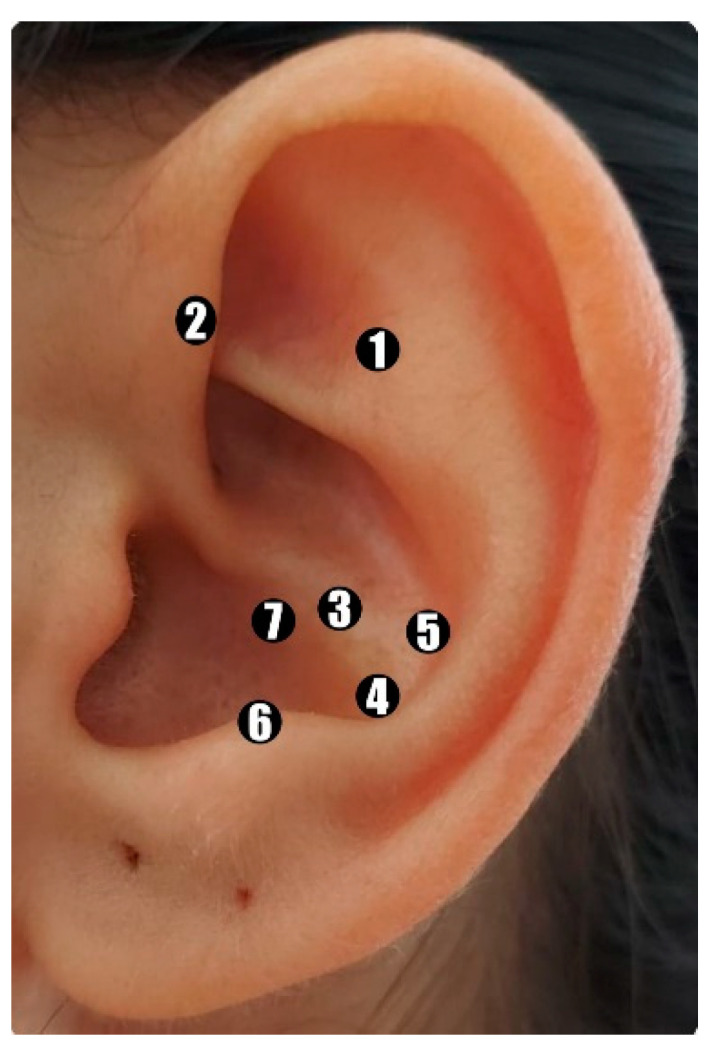
Representation of the auricular points used to manage CINV-Points located according to the map of the World Federation of Acupuncture-Moxibustion Societies [21]: 1-Shénmén (TF4) shenmen; 2-Jiaogan (AH6a) sympathetic nerve (internal point); 3-Wèi (CO4) stomach; 4-Pí (CO13) spleen; 5-Gän (CO12) liver; 6-Pízhìxià (AT4) subcortex; 7-bënmén (C03) cardia.

**Table 1 healthcare-12-00218-t001:** Characteristics of the panel specialists.

Variables	*n* (%)
Sex	
Female	12 (85.7)
Male	2 (14.3)
Time of practical experience	
Between 4 and 5 years	4 (28.6)
Between 6 and 10 years	4 (28.6)
Between 11 and 15 years	2 (14.3)
Between 16 and 24 years	2 (14.3)
Over 25 years	2 (14.3)
Time of teaching experience	
No experience	5 (35.7)
Between 3 and 5 years	2 (14.3)
Between 6 and 10 years	3 (21.4)
Between 11 and 15 years	2 (14.3)
Between 16 and 24 years	2 (14.3)
Research experience with published articles on auricular acupuncture (AA) and/or acupuncture in relevant journals	
Yes	13 (92.9)
No	1 (7.1)
Participation in research groups in the field of AA and/or acupuncture	
Yes	13 (92.9)
No	1 (7.1)
Master’s degree with a dissertation in AA and/or acupuncture	
Yes	7 (50.0)
No	7 (50.0)
PhD with a thesis in the field of AA and/or acupuncture	
Yes	5 (35.7)
No	9 (64.3)
Course/specialization or residency in the field of AA and/or acupuncture	
Yes	13 (92.0)
No	1 (7.1)

**Table 2 healthcare-12-00218-t002:** Final version and assessment of the content validity of the AA protocol for CINV management (number of specialists = 14).

Item Description	Final Version of the Proposed Protocol	Agreement Index (%)
Acupuncture style	Auricular acupuncture (AA) based on the precepts of traditional Chinese medicine (TCM)	92.9
Reasoning for treatment provided, based on historical context, literature sources, and/or consensus methods, with references where suitable	AA is a specialized approach to TCM/acupuncture which uses the auricular microsystem for the treatment, prevention, and diagnosis of health conditions [36]. It is currently used in more than 249 areas of the health sciences, including oncology [36]. AA’s physiological mechanism of action is still under study [14]. However, it is believed to be neurologically based on stimulation of the trigeminal and vagus nerves, thus generating neuromodulation in the central nervous system (CNS). In addition, the stimulation of auricular points may be related to an increase in vagal tone, as well as regulations in the gastrointestinal, endocrine, cardiovascular, and respiratory systems [14,15].	85.7
Extent to which treatment may vary	All patients will receive the same treatment protocol in all five sessions. The final version of the protocol will be based on the results of a previous systematic review [17], on content validation by specialists in the field, and on clinical validation.	85.7
Number of device insertions per subject and per session	Seven auricular seeds will be applied to each subject per session.	100
Names (or location if no standard name) of points used (uni-/bilateral)	Shénmén (TF4) shenmen, Jiaogan (AH6a) sympathetic nerve, Wèi (CO4) stomach, Pí (CO13) spleen, Gän (CO12) liver, Pízhìxià (AT4) subcortex, and bënmén (C03) cardia. These points will be applied unilaterally, alternating ears each session.	100
Depth of insertion	Not applicable.	100
Response sought	Patients will be instructed to press the seeds at three different times, in the morning (3 times), in the afternoon (3 times), and in the evening (3 times), and whenever they feel nauseous or vomit, for approximately 30 s at each point, until the ear becomes slightly hyperemic [39] or until they feel slight discomfort or pain [29,37]—with a sensation of “deqi”.	100
Device stimulation	Manual seed acupressure.	92.9
Device retention time	Patients will be instructed to keep the seeds fixed in their ears for a period of seven days. In the event that they are removed before this period (intentionally or unintentionally), patients will be instructed to inform the researcher.	92.9
Device type	Vaccaria seeds	85.7
Number of treatment sessions	Five sessions	92.9
Frequency and duration of treatment sessions	Once a week, for approximately 20 min each session. The entire treatment will last approximately five weeks.	100
Details of the other interventions administered	No other interventions will be applied.	100
Setting and context of treatment, including instructions to practitioners, and information and explanations to patients	Only a qualified interventionist will apply the intervention. After attaching the seeds, participants will be instructed to press each point for 30 s, three times a day, in the morning, afternoon, and evening, and whenever they feel nauseous or vomit [38]. In addition, they will be instructed to keep the devices inserted until the next AA session, for a period of seven days, or to remove them in the event of discomfort, allergic processes, and/or itching [40]. They will also be instructed on how to maintain hygiene and how to keep the seeds in the ear, as well as how to address the possible discomfort resulting from the procedure. These guidelines will be reinforced with each new session. To reinforce the necessary care, patients will receive a leaflet containing guidelines on how to maintain the stitches, the scheduled days for the AA sessions and evaluations, as well as a contact number of the researcher in charge.	92.90
Description of participating acupuncturists	Academic qualifications: Nurse. Specialization in classical systemic acupuncture, with 1200 h/Minimum of two years of experience; 80 h AA course/Minimum of four years of experience.	100

**Table 3 healthcare-12-00218-t003:** Participants’ demographic and clinical characteristics (*n* = 20).

Variables	*n* (%)
Sex	
Male	5 (25)
Female	15 (75)
Marital status	
Single	4 (20)
Married	11 (55)
Divorced	2 (10)
Widowed	3 (15)
Race	
White	18 (90)
Black	2 (10)
Frequency of chemotherapy sessions	
Weekly	9 (45)
Fortnightly	4 (20)
Three weeks and a weekly break	4 (20)
21 days	3 (15)
Duration of chemotherapy treatment	
Less than six months	9 (45)
Between six months and one year	3 (15)
Between one and six years	2 (10)
Over three years	6 (30)
Type of cancer	
Breast	4 (20)
Multiple myeloma	4 (20)
Colon	2 (10)
Prostate	3 (15)
Esophagus	2 (10)
Ovary	2 (10)
Rectum	2 (10)
Stomach	1 (5)
Chemotherapy protocol	
Carboplatin and paclitaxel	4 (20)
Folfiri (fluoruracil, calcium folinate, and irinotecan)	3 (15)
Bortezomib	2 (10)
Nordic flox (5-fluorouracil, leucovorin combined with oxaliplatin)	1 (5)
Doxorubicin and cyclophosphamide	2 (10)
Folfoxiri (folinic acid, 5-fluorouracil, oxaliplatin and irinotecan)	1 (5)
Gemcitabine	2 (10)
Cisplatin, vincriatin, and filgrastim	1 (5)
Leuprorelin combined with bicalutamide prostate and with cyclophosphamide and leuprorelin	1 (5)
Leuprorelin and zoledronic acid	1 (5)
Cisplatin and Gemcitabine	1 (5)
Vinorelbine and trastuzumab	1 (5)

**Table 4 healthcare-12-00218-t004:** CINV assessment of study participants before and after applying AA (*n* = 20).

Variables	Pretest	Post-Test	*p*
	*n* (%)	*n* (%)	
Presented nausea after the last chemotherapy session	20 (100)	11 (55)	0.004 *
How often have you felt nauseous after the chemotherapy session?			
Never	0	9 (45)	0.001 **
1 to 3 times	6 (30)	9 (45)	
4 to 6 times	3 (15)	0	
7 to 9 times	0	0	
9 times or more	11 (55)	2 (10)	
How was your worst nausea feeling after your last chemotherapy session?			
I did not feel nauseous	0	9 (45)	0.001 **
Very light	0	2 (10)	
Light	2 (10)	4 (20)	
Moderate	3 (15)	2 (10)	
Strong	8 (40)	2 (10)	
Very strong	6 (30)	1 (5)	
Unbearable	1 (5)	0	
When was your worst nausea feeling?			
I did not feel nauseous	0	9 (45)	0.002 **
During chemotherapy	1 (5)	0	
0 to 4 h after chemotherapy	3 (15)	2 (10)	
5 to 8 h after chemotherapy	2 (10)	0	
9 to 12 h after chemotherapy	0	0	
13 to 24 h after chemotherapy	0	0	
Over 24 h after chemotherapy	9 (45)	7 (35)	
The feeling of nausea remained the same the entire time	5 (25)	2 (10)	
Experienced vomiting after the last chemotherapy session	13 (65)	7 (35.0)	0.031 *
How many times have you vomited after the chemotherapy session?			
Never	6 (30)	13 (65)	0.007 **
1 to 3 times	10 (50)	5 (25)	
4 to 6 times	1 (5)	1 (5)	
7 to 9 times	1 (5)	0	
9 times or more	2 (10)	1 (5)	
What was the worst instance of vomiting you have experienced?			
I did not vomit	6 (30)	13 (65)	0.052 **
Very light	0	1 (5)	
Light	4 (20)	2 (10)	
Moderate	3 (15)	0	
Strong	3 (15)	3 (15)	
Very strong	4 (20)	1 (5)	
Unbearable	0	0	
When was the worst instance of vomiting you have experienced?			
I did not vomit	6 (30)	12 (60)	0.021 **
During chemotherapy	0	0	
0 to 4 h after chemotherapy	1 (5)	2 (10)	
5 to 8 h after chemotherapy	0 (0)	0	
9 to 12 h after chemotherapy	0 (0)	0	
13 to 24 h after chemotherapy	1 (5)	0	
Over 24 h after chemotherapy	10 (50)	5 (25)	
The feeling of vomiting remained the same the entire time	2 (10)	1 (5)	
Experienced nausea before chemotherapy?	7 (35)	3 (15)	0.219 *
How was your feeling of nausea before chemotherapy?			
I did not feel nauseous	13 (65)	17 (85)	0.034 **
Very light	0	0	
Light	1 (5)	3 (15)	
Moderate	0	0	
Strong	2 (10)	0	
Very strong	3 (15)	0	
Unbearable	1 (5)	0	
How long before your chemotherapy session did you feel nauseous?			
I did not feel nauseous before the last chemotherapy session	13 (65)	17 (85)	0.059 **
I felt nauseous 1 to 3 h before the last chemotherapy session	1 (5)	2 (10)	
I felt nauseous 4 to 6 h before the last chemotherapy session	1 (5)	0	
I felt nauseous 7 to 9 h before the last chemotherapy session	0	0	
I felt nauseous 9 h before the last chemotherapy session	5 (25)	1 (5)	
Vomited before the last chemotherapy session	3 (15)	0	0.250 *
What was the worst vomiting experience you had before your last chemotherapy session?			
I did not vomit	17 (85)	20 (100)	0.083 **
Very light	0	0	
Light	1 (5)	0	
Moderate	0	0	
Strong	1 (5)	0	
Very strong	1 (5)	0	
Unbearable	0 (0)	0	
How long before your last chemotherapy session did you vomit?			
I did not vomit before the last chemotherapy session	18 (90)	20 (100)	0.157 **
I vomited 1 to 3 h before the last chemotherapy session	1 (5)	0	
I vomited 4 to 6 h before the last chemotherapy session	0 (0)	0	
I vomited 7 to 9 h before the last chemotherapy session	0 (0)	0	
I vomited 9 h before the last chemotherapy session	1 (0)	0	
Have you taken any medication for nausea and/or vomiting after your last chemotherapy session?	16 (80)	15 (75)	1.000 *
Did this medication help?			
Yes	10 (50)	11 (55)	0.405 **
A little	1 (5)	3 (15)	
Very little	2 (10)	0	
No	3 (15)	2 (10)	
I did not use any medication	4 (20)	4 (20)	

* Exact significances are displayed for the McNemar test. ** Exact significances are displayed for the Friedman test.

## Data Availability

All relevant data are included in this manuscript and Appendix A.

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
