# Peer review of "Development and Validation of an Auricular Acupuncture Protocol for the Management of Chemotherapy-Induced Nausea and Vomiting in Cancer Patients"

_healthcare, 2024, doi:10.3390/healthcare12020218_

Round 1

Reviewer 1 Report

Comments and Suggestions for Authors

This is a very well-written study with a good description of the study protocol and a very impressive description of the characteristics of the panel experts. Results are valid based on the small sample size. 

Line 160-suggest you add exclusion criteria included ....

No recommended sample size calculated?

I recommend adding to the limitations that there was a small sample size. 

As a reader, I would like to see a picture of the AA seeds, but appreciate the figure of the auricular points. 

Comments on the Quality of English Language

Be careful of capitalizing words that do not require it, such as in line 60 and line 76. McNemar's test is written with no space. Good command of the English language. 

Author Response

We would like to thank you and the reviewers for the comments provided on our manuscript. Below, we provide a point-by-point response to the reviewers’ comments (responses in bold italics following Author Response ‘AR’). We herewith resubmit our revised manuscript as a Review article for consideration for publication in Healthcare Journal.

Reviewer 2 Report

Comments and Suggestions for Authors

Line 148 - Would be helpful to know the rationale for exclusion of ECOG 3+ (I assume this is related to fitness for chemotherapy, but having this explicitly stated would be helpful)

Line 166 - an explanation of what 'obstructive' refers to would be helpful - would this be related to obstruction of the ear canal? Obstruction of the protocol?

I was unable to find the time between each session of AA until getting to the table - would be helpful to include in the text. Also, was there any consideration to how this (and MANE measurement) may line up according to chemotherapy cycle and the typical temporal sequence of CINV on a given protocol?

Table 2 - why was marital status and race captured? Was this hypothesized to influence outcomes? In the Brazilian context, is race typically captured as white or black (I may be unfamiliar - this would be captured very differently in the Canadian context and such a narrow definition of race would not even likely be approved for collection by an REB locally, though I appreciate this is likely specific to jurisdiction). Additionally, I feel the use of the term 'color' is not appropriately sensitive. 

Table 2 - Was the information about the type of chemotherapy (e.g. targeted therapies, platinum based chemo, etc.) captured? This would be very interesting to know, though perhaps beyond the scope of this protocol but may be worth mentioning as part of limitations. 

Author Response

(The authors gave the same response as above.)

Reviewer 3 Report

Comments and Suggestions for Authors

The study was designed on a very interesting subject and has a high potential to contribute to the literature, I would like to congratulate the authors. There are some specific issues that need to be addressed before the study can be considered appropriate for journal, however;

1. The study population is quite small, which makes it difficult to evaluate the results statistically. Have you done a power analysis of the population and I would like to see the results.

2. You also referenced similar works of yours in the references section. State the differences between your previous works and this work.

3. A recent review study similar to this study was conducted. (Paiva, E. M. C.; Zhu, S.; Chi, Y.; Oliveira, R. A.; Moura, C. C.; Garcia, A. C. M. Auriculotherapy to manage chemotherapy-induced 448 nausea and vomiting in patients with cancer: a systematic review, Progress in Palliative Care, 2023, 31, 100-110). They also added this study to the references section. They have another study on the same subject, in which they recently published an evaluation scale. This is also in the references section. ( Isidoro, G. M.; Ferreira, A. C. G.; Paiva, E. M. C.; Amaral, J. D. H. F.; Meireles, E. C. A.; Garcia, A. C. M. Escala para Avaliação de 500 Náuseas e Vômitos Relacionados à Quimioterapia: Tradução e Adaptação Transcultural. RBC, 2022, 68, 1-11).  Frankly, it's a bit confusing. Authors should be asked for information on this subject.

Author Response

(The authors gave the same response as above.)

Reviewer 4 Report

Comments and Suggestions for Authors

My report for the above manuscript is:

This paper seeks to develop a protocol to quantify the role of acupuncture in the management of side effects of chemotherapy. Using a two-stage process, including a literature review and pilot study, the authors develop an instrument which they argue can be used for a larger trial to assess its value for the target audience.

Overall, the paper is very well constructed and argued. The authors have provided a logical rationale for their work, and the approach sound. There are some minor grammar changes required and the authors may like to consider some small changes to help an audience with limited knowledge of acupuncture to appreciate the validity of their approach, not least given the wider specialities involved in the management of cancer care.

These developments include:

The selection of 14 specialists. The term specialist includes a wide range of attributes which while having been described may remain vague. For example a PhD could be in a variety of associated fields that have limited connection to their present roles. The current position of the specialist may be of greater value.

Associated with is the question of why 14? Is there some rationale for the number?

Some addition background information of the “World Federation of Acupuncture-moxibustion societies” may also be helpful.

Overall though the authors have undertaken and presented a sound piece of work.

Author Response

(The authors gave the same response as above.)
